# Evidence for Non-Cancer-Specific T Cell Exhaustion in the Tcl1 Mouse Model for Chronic Lymphocytic Leukemia

**DOI:** 10.3390/ijms22136648

**Published:** 2021-06-22

**Authors:** Thomas Parigger, Franz Josef Gassner, Christian Scherhäufl, Aryunni Abu Bakar, Jan Philip Höpner, Alexandra Hödlmoser, Markus Steiner, Kemal Catakovic, Roland Geisberger, Richard Greil, Nadja Zaborsky

**Affiliations:** 1Department of Internal Medicine III with Haematology, Medical Oncology, Haemostaseology, Infectiology and Rheumatology, Oncologic Center, Salzburg Cancer Research Institute—Laboratory for Immunological and Molecular Cancer Research (LIMCR), Paracelsus Medical University, 5020 Salzburg, Austria; t.parigger@salk.at (T.P.); f.gassner@salk.at (F.J.G.); c.scherhaeufl@salk.at (C.S.); a.abu-bakar@salk.at (A.A.B.); j.hoepner@salk.at (J.P.H.); a.hoedlmoser@salk.at (A.H.); mark.steiner@salk.at (M.S.); kemal.catakovic@abbvie.com (K.C.); r.greil@salk.at (R.G.); 2Department of Biosciences, Paris-Lodron-University Salzburg, 5020 Salzburg, Austria

**Keywords:** CLL, T cell exhaustion, Tcl1

## Abstract

The reinvigoration of anti-cancer immunity by immune checkpoint therapies has greatly improved cancer treatment. In chronic lymphocytic leukemia (CLL), patients as well as in the Tcl1 mouse model for CLL, PD1-expressing, exhausted T cells significantly expand alongside CLL development; nevertheless, PD1 inhibition has no clinical benefit. Hence, exhausted T cells are either not activatable by simple PD1 blocking in CLL and/or only an insufficient number of exhausted T cells are CLL-specific. In this study, we examined the latter hypothesis by exploiting the Tcl1 transgenic CLL mouse model in combination with TCR transgene expression specific for a non-cancer antigen. Following CLL tumor development, increased PD1 levels were detected on non-CLL specific T cells that seem dependent on the presence of (tumor-) antigen-specific T cells. Transcriptome analysis confirmed a similar exhaustion phenotype of non-CLL specific and endogenous PD1pos T cells. Our results indicate that in the CLL mouse model, a substantial fraction of non-CLL specific T cells becomes exhausted during disease progression in a bystander effect. These findings have important implications for the general efficacy assessment of immune checkpoint therapies in CLL.

## 1. Introduction

The acquisition of cancer-specific mutations and splicing patterns normally leads to the expression of cancer-specific antigens and hence, to a cancer-specific immunopeptidome presented on MHC molecules [1,2,3]. Although this should result in the recognition of cancer cells by the host’s immune system, a robust anti-cancer immune response is frequently rendered ineffective due to many immunosuppressive mechanisms from the cancer cells and the cancer microenvironment [4]. A prominent T cell dysfunction associated with cancer is T cell exhaustion [5]. T cell exhaustion was originally discovered in mouse models investigating chronic, persisting virus infections. Within such chronic settings, virus-specific T cells, characterized by the expression of the inhibitory receptor PD1, are silenced to avoid overwhelming tissue destruction and immune pathologies [6]. The primary cause for the exhaustion of T cells is repetitive antigen recognition and therefore TCR stimulation with simultaneous inhibitory signals coming from the microenvironment [6,7,8]. Analogously, increased T cell exhaustion was found in cancer patients and the reactivation of exhausted T cells using immune checkpoint inhibitors (ICI) such as PD1 blocking antibodies, proved to have surprising clinical benefits for some cancer patients [9]. However, many cancer patients, particularly chronic lymphocytic leukemia (CLL) patients, do not respond to ICI therapy for unknown reasons [10]. Previous studies have revealed the substantial induction of PD1 expression in T cells from CLL patients and Tcl1 mice alongside CLL development and progression [11,12]. Nevertheless, the extent of T cells with CLL specificity vs. non-specificity has not been determined so far, albeit it is likely decisive for the efficacy of ICI therapy. Thus, the aim of this study is to elucidate whether T cell exhaustion is specifically confined to tumor-specific T cells or whether this dysfunctional phenotype also affects T cells independently of recognizing tumor antigens.

As CLL specificity is hard to assess when the repertoire of MHC-I/MHC-II-presented CLL antigens is unknown, we aimed at investigating PD1 expression in T cells specific for an irrelevant antigen, which are hence referred to as non-CLL-specific. We did this by using the Tcl1 mouse transplant model for CLL [13] and congenic OT1 and OT2 mice [14,15] as recipients for transferred Tcl1 tumors. OT1/OT2 mice express a transgenic TCR on CD8 T cells (OT1) or CD4 T cells (OT2) that are specific for ovalbumin-derived peptides presented on MHC-I or MHC-II molecules. As the transferred Tcl1 CLL cells as well as the OT1/OT2 recipients do not express ovalbumin, the transgenic T cells constitute a bona-fide non-CLL-specific T cell population. While OT1/OT2 mice generate T cells, of which the vast majority expresses the OVA-specific TCR, only a small percentage of cells express a rearranged endogenous TCR corresponding to a diverse WT T cell repertoire. However, OT1/OT2 mice on a *Rag*-deficient background (OT1 Rag/OT2 Rag) exclusively have T cells expressing the transgenic TCR, as *Rag* is necessary for the productive rearrangement of the endogenous TCR [16,17]. Strikingly, our study indicates that CLL cells are able to induce T cell exhaustion in an antigen-independent manner in non-CLL-specific CD4 and CD8 T cells. This bystander T cell exhaustion was augmented in the presence of T cells with a WT TCR repertoire, as this effect was less pronounced in mice on a Rag-deficient background. Our data indicate that simple PD1 analyses on T cells in the context of cancer does not allow conclusions for a possible reactivation of cancer-specific T cells using ICI and reveals substantial bystander exhaustion in T cells during CLL development and possibly in other cancers as well.

## 2. Results

### 2.1. PD1 Expression Is Upregulated in Non-CLL-Specific T Cells upon Leukemia Development

First, we compared the induction of PD1 upregulation on T cells as a surrogate marker for T cell exhaustion upon transplantation of CLL cells from the Tcl1 CLL mouse model into the congenic WT and OT1/OT2 mice as recipients (Figure 1A). OT1 and OT2 mice possess a fully rearranged transgenic TCR, highly specific for an ovalbumin-derived peptide restricted to either MHC-I (OT1) or MHC-II (OT2) (Appendix A, left panel). As ovalbumin is not expressed in our mouse models, any T cells expressing the transgenic TCR, which are easily identifiable by staining with a TCR-Vb5-specific antibody, were defined as non-CLL-specific. As only a small percentage of endogenous Vb5pos T cells is present in WT mice (5.0% Vb5.1pos/5.2pos T cells (SD = 0.5; *n* = 6); [18]), only a small fraction of this Vb5pos population could potentially be tumor-specific within the OT mouse model. In line with our previous report [11], we observed an induction of PD1 expression in CD4 as well as CD8 T cells in peripheral blood samples during disease development in CLL-transplanted congenic WT recipient mice (mean CD4: preTx 11.7%, postTx 23.2%, *p* = 0.0091, *n* = 14; mean CD8: preTx 3.6%, postTx 16.8%, *p* = 0.0002, *n* = 14) (Figure 1B,C). However, OT1 and OT2 recipient mice also exhibited robust PD1 expression in non-CLL-specific Vb5pos T cells, similar to the extent observed in CD4/CD8 T cells from WT recipients (mean OT2: preTx 2.3%, postTx 10.8%, *p* = 0.0049, *n* = 16; mean OT1: preTx 1%, postTx 13.5%, *p* = 0.0022, *n* = 16) (Figure 1B,C). Comparing CLL developments revealed that OT1 and OT2 recipients had accelerated leukemia development compared to WT recipients (50% TL: OT1 mean 30.6d, SD 7.7d, *n* = 16; OT2 mean 29.5d, SD 3d, *n* = 16; WT mean 32.3, SD 8.3, *n* = 12), resulting in a shortened overall survival (OS), particularly for OT1 recipients (OS: OT1 mean 31.2d, SD 6.7d, *n* = 16; OT2 mean 37d, SD 8.1d, *n* = 16; WT mean 42.7d, SD 7.7d, *n* = 14) (Appendix A).

### 2.2. PD1 Upregulation on Non-CLL-Specific T Cells Is Independent of Endogenous TCR Expression and Correlates with PD1 Levels of Endogenous T Cells

Next, we examined whether T cells expressing the transgenic TCR co-express a second endogenous TCR, as this could result in bispecific T cells, possibly recognizing CLL-derived antigens via the endogenous TCR and thereby inducing PD1 expression. Therefore, we determined the level of T cells co-expressing a Vb5-TCR together with a TCR that consists of another TCR Vb family. We stained splenocytes from OT1 and OT2 tumor-recipient mice using a Vb5.1 + Vb5.2-specific PE-labelled antibody and a cocktail of FITC-conjugated antibodies specific for all commercially available TCR Vb chains (VbX) (Figure 2A), whereby about 71% (SD 3.9, *n* = 6) of the endogenous TCR repertoire of WT mice is covered [18]. We observed that >90% of the CD8+ T cells from OT1 mice expressed only the transgenic Vb5-TCR, whereas in OT2 mice about half of the CD4+ T cell compartment was single-positive for Vb5. The double positive T cell fraction, which co-expresses the transgenic TCR together with an endogenous TCR, was found to be almost non-existent in OT1 mice (mean 0.80%; SD 0.25; *n* = 3) and low in OT2 mice (mean 4.48%; SD = 1.42; *n* = 3) (Figure 2B). However, the Vb5 single-positive T cells showed a robust PD1 positivity in about 10.17% (SD 4.57; *n* = 3) of the CD8pos T cell compartment in OT1 mice and in about 47.31% (SD 9.70; *n* = 3) of CD4pos T cells of OT2 mice, suggesting a mechanism for PD1 upregulation independent from cognate TCR–MHC interaction. Notably, the percentage of PD1pos cells was higher in T cells expressing an (additional) endogenous TCRs (OT2: VbXpos 79.9%, SD 4.3%; VbXposVb5pos 78.3%, SD 1.3%, *n* = 3; OT1: VbXpos n.d., VbXposVb5pos n.d.) (either covered or not covered by the VbX antibody cocktail, reflected in VbX positivity or Vb5/VbX double negativity), showing that TCR specificity is still a major driver for PD1 expression in T cells (Figure 2B). Additionally, we noticed that the presence of PD1pos T cells among the residual Vb5neg fraction, constituting the residual normal T cell repertoire, correlated with PD1pos Vb5pos T cells, although this correlation was more pronounced and statistically significant only in OT2 compared to OT1 mice (OT1: R2 0.23, *p* = 0.1; OT2 R2 0.67, *p* = 0.0003) (Figure 2C). This could indicate that normal T cells—possibly also comprising some CLL-specific T cells—are necessary to induce PD1 expression in non-CLL-specific Vb5pos T cells via a bystander effect. In conclusion, the co-expression of endogenous TCR cannot be the driving factor of PD1 expression in non-cancer-specific T cells.

### 2.3. PD1 Expression of Non-CLL-Specific T Cells Is Dependent on the Presence of Potentially Tumor-Specific T Cells

To examine whether PD1pos non-CLL-specific T cells are generated in the absence of any residual normal T cell repertoire, we used OT1/OT2 mice on a *Rag* knockout background (OT1 Rag/OT2 Rag mice) as recipients. In these mice, endogenous TCR is not expressed, as its rearrangement is dependent on the Rag recombinase. As a result, these mice only possess T cells exclusively expressing the transgenic, ovalbumin-specific TCR. Consequently, all the T cells from the Rag knockout OT1/OT2 mice were Vb5pos, which means that these mice only have OVA-specific (and thus non-CLL-specific) T cells (Appendix A, right panel). In order to avoid transferring high numbers of WT T cells from spleen samples into OT Rag mice, CLL cells were purified from splenocytes derived from two transplanted (Tx) WT mice (WT 371 Tx and WT 370 Tx) by magnetic-activated cell sorting (MACS) and were subsequently transplanted individually into Rag-deficient and -proficient OT mice and WT mice, respectively. As a control for CLL-independent processes of PD1 regulation, healthy B cells of an untransplanted (no Tx) WT mouse (483) were simultaneously isolated via MACS and transplanted into OT1 Rag mice (Figure 3A). However, there were still residual T cells (1.7–3.5% CD3pos T cells) detected in the T cell-depleted CLL samples (Appendix A). As expected, PD1 levels in spleen samples of transplanted WT mice increased significantly in the CD4 and CD8 T cell compartment upon tumor injection compared to untransplanted mice (CD8 Tx: mean 42.3%, SD 18.1%, *n* = 12; CD8 noTx: mean 4.2%, SD 2.4%, *n* = 3, *p* = 0.003; CD4 Tx: mean 45.5%, SD 16.5%, *n* = 12; CD4 noTx: mean 20.3%, SD 9.2%, *n* = 3, *p* = 0.02) (Figure 3B,C). Similarly, we again detected increased PD1 expression on OVA-specific T cells (Vb5pos) in spleen samples of transplanted versus non-transplanted OT1 and OT2 mice (OT1 Tx: mean 16.8%, SD 10.8%, *n* = 12; CD8 noTx: mean 0.1%, SD 0.06%, *n* = 3, *p* = 0.015; OT2 Tx: mean 24.3%, SD 16.9%, *n* = 14; CD4 noTx: mean 3.2%, SD 2.7%, *n* = 3, *p* = 0.053) (Figure 3B,C).

Strikingly, upon transfer of purified CLL cells into OT1 Rag and OT2 Rag mice, the aforementioned upregulation of PD1 on OVA-specific Vb5pos T cells was low or completely absent in peripheral blood samples of OT2 Rag and OT1 Rag mice, respectively (OT1 Rag: preTx mean 0.1%, SD 0.04%, postTx mean 0.3%, SD 0.3%, *n* = 5, *p* = 0.122; OT2 Rag: preTx mean 0.59%, SD 0.2, postTx mean 4.4%, SD 2.4%, *n* = 4, *p* = 0.0485) (Figure 3D,E), compared to the previously shown data on Rag-proficient OT mice (mean OT1: preTx 1%, postTx 13.5%, *p* = 0.0022, *n* = 16; mean OT2: preTx 2.3%, postTx 10.8%, *p* = 0.0049, *n* = 16;) (Figure 1B). Similarly, we found a significantly lower PD1 expression on splenic CD8+ T cells in OT1 Rag mice compared to conventional OT1 mice (OT1 Rag: mean 4.9%, SD 2.6%, *n* = 5; OT1: mean 16.8%, SD 10.8%, *n* = 12; *p* = 0.02) (Figure 3B,H left panel). However, spleen-derived Vb5pos T cells in OT2 Rag mice did not show a significant reduction of PD1 expression on CD4+ T cells compared to Rag-proficient OT2 mice (OT2 Rag: mean 26.8%, SD 22.6%, *n* = 4; OT2: mean 24.3%, SD 19.9%, *n* = 14, *p* = 0.812) (Figure 3C,H right panel). Notably, in OT2 Rag mice we detected a decrease of OVA-specific T cells (Vb5pos) which goes along with an increase of endogenous (Vb5neg) T cells during disease development in the peripheral blood (Appendix A, lower left panel). However, no reduction of the Vb5pos T cell fraction was detected in OT1 Rag mice (Vb5pos T cell fraction: OT1 Rag mean 96.6%, SD = 3.9%, *n* = 5; OT2 Rag mean 74.6%, SD = 26.1%, *n* = 4; *p* = 0.1) (Appendix A, lower right panel). Interestingly, we observed an increase of endogenous T cells (Vb5neg) in spleen samples of transplanted OT2 Rag mice and in transplanted OT1 Rag mice (Vb5neg T cell fraction: OT1 Rag Tx 4.9%, SD = 2.6, *n* = 5; OT2 Rag Tx 56.6%, SD = 16.8, *n* = 4; *p* = 0.001, Appendix A).

The massive reduction of PD1+ CD8+ T cells in OT1 Rag Tx compared to OT1 Tx and WT Tx mice significantly correlated with a low amount of endogenous T cells (Vb5neg fraction: OT1 Rag mean 4.9%, SD 2.6%, *n* = 5; OT1 mean 24.3%, SD 12.9%, *n* = 12; WT mean 91.5%, SD 1.8%, *n* = 12) (Figure 3F). In the OT2 Rag Tx, OT2 Tx, and WT Tx mice, we found a similar highly significant correlation of the percentage of PD1+ CD4+ T cells with the amount of endogenous Vb5neg T cells (Vb5neg fraction: OT2 Rag mean 56.6%, SD 16.8%, *n* = 4; OT2 mean 35%, SD 17.3%, *n* = 12; WT mean 91.5%, SD 1.8%, *n* = 12) (Figure 3G). This suggests an implication of potentially tumor-reactive T cells (in the Vb5neg T cell fraction) in conveying PD1 expression to OVA-specific T cells.

Finally, CLL development determined from consecutive blood draws revealed accelerated tumor development in OT2 Rag mice and shortened overall survival in OT1 Rag and OT2 Rag mice as compared to WT recipients (50% TL: OT1 Rag mean 33.2d, SD 8.9d, *n* = 5, OT2 Rag mean 26.2d, SD 3.5d, *n* = 4, WT mean 32.2d, SD 8.3d, *n* = 12; OS: OT1 Rag mean 30.2d, SD 14.4d, *n* = 5, OT2 Rag mean 27.5d, SD 1d, *n* = 4, WT mean 42.7d, SD 7.7d, *n* = 14) (Appendix A). BCR analysis of CLL tumors as well as transplanted tumors in recipient mice revealed no clonal shifts upon tumor transfer in any mouse genotype, except for one transplanted OT1 mouse (ID 423), in which the clonal composition changed, with a previously minor clone expanding. (Appendix A).

### 2.4. Transcriptome of PD1pos Non-CLL-Specific T Cells Mimics Expression Signature of Functionally Exhausted T Cells

Subsequently, we wanted to determine whether PD1pos T cells differ from the PD1neg T cell population on a transcriptional level and whether they exhibit an expression signature that resembles exhausted T cells, as previously published by Crawford et al. [19]. Therefore, we extracted RNA from purified Vb5pos PD1pos and Vb5pos PD1neg CD4 T cells isolated from OT2 Tx recipient mice at tumor endpoint and analogously, from purified Vb5pos PD1pos and Vb5pos PD1neg CD8 T cells isolated from OT1 Tx recipient mice at tumor endpoint. Additionally, we sorted PD1pos and PD1neg CD8/CD4 T cells from WT Tx mice at endpoints (Appendix A). Purities of sorted cell populations ranged from 97–100% (data not shown). Gene expression profiling of sorted CD8 PD1pos vs. PD1neg T cells revealed many deregulated genes belonging to the exhaustion gene set previously defined by Crawford et al. [19] (Figure 4A and Appendix A). However, we could not detect differentially expressed genes between PD1pos CD8 T cells from WT mice compared to Vb5pos PD1pos T cells of OT1 mice (Figure 4A right panel), corroborating that the PD1 expressing subsets from WT and OT1 mice possess similar expression profiles that resemble the previously defined exhaustion phenotype [19]. Differential expression analysis in the CD4 T cell compartment displayed a less pronounced gene expression difference between PD1pos and PD1neg subsets of WT mice. Nevertheless, we found significantly deregulated CD4-specific exhaustion markers such as *IL-21*, *Fosb*, *Dusp14*, and *Plk2* in the Vb5pos PD1pos T cells of OT2 mice. Again, no differential gene expression was observed in the subset of PD1pos CD4 T cells of WT vs. Vb5pos PD1pos CD4 T cells of OT2 T mice (Figure 4A lower right panel). In order to further verify that our sorted PD1 expressing T cells truly represent an exhausted phenotype, we applied an exhaustion gene set defined by Crawford et al. for CD4 (comprising 491 genes) and CD8 T cells (comprising 608 genes) [19] in a gene set enrichment analysis (GSEA). Notably, this analysis revealed that the PD1pos CD4 and CD8 T cells of transplanted WT, as well as the PD1pos Vb5pos T cells of OT1 Tx and OT2 Tx mice, indeed show a highly significant exhaustion signature (Figure 4B).

## 3. Discussion

Immune checkpoint inhibition (ICI) to reactivate anti-cancer immunity is currently one of the most promising therapeutic approaches. However, in many cancers such as CLL, patients do not show any clinical benefits upon ICI treatment [20], whereas predictive markers and mechanisms that contribute to ICI efficacy are important research topics. We and others have previously shown that T cell exhaustion, as reflected by expression of inhibitory receptors such as PD1, increases during CLL development in patients as well as in the Tcl1 mouse model [11,12]. T cell exhaustion is mainly driven by repetitive recognition of antigens, and in the context of cancer, tumor-specific neoantigens [7]. Therefore, it is conclusive that tumors with a high mutational burden, such as melanoma, show strong T cell exhaustion phenotypes and a good response towards ICI therapies. In contrast, CLL possesses a low mutational burden and thus probably a low number of putative neoantigens, which could trigger T cell exhaustion. Nevertheless, a substantial fraction of T cells is highly dysfunctional in CLL, raising the question about the mechanism of action which drives the formation of T cell exhaustion in CLL. Therefore, we assumed that a large extent of these exhausted T cells could be non-CLL specific. An ICI treatment based solely on the presence of exhausted T cells might therefore unintentionally target primarily non-CLL T cells, hence rendering any potential non-CLL T cell activation clinically ineffective. Furthermore, a potential induction of T cell exhaustion via a neoantigen-independent mechanism would create a comprehensive immune-suppressive environment, possibly making ICI, which targets only one component of this suppressive environment, insufficient for the reactivation of potentially tumor-specific T cells. In this study, we examined this hypothesis and found that in CLL tumors derived from the Tcl1 mouse model, non-CLL-specific T cells in OT1/OT2 recipient mice indeed became PD1 positive as observed in T cells with a normal TCR repertoire. This PD1 positivity within non-CLL-specific T cells was clearly associated with an exhaustion signature, which did not differ from the signature of WT T cells potentially including cancer-specific T cells. These data suggest that analogously, a substantial fraction of exhausted T cells observed in CLL patients may be unspecific towards the malignant cells and were likely instructed to become exhausted in a bystander effect, independent of antigen-specificity. This bystander effect must be partially dependent on a normal T cell repertoire, as OVA-specific T cells displayed higher PD1pos fractions when residual WT T cells with a diverse TCR repertoire were present. This is reflected in higher exhaustion levels in OT1/OT2 mice as compared to Rag-deficient OT1/OT2 mice and in the correlation of higher PD1 levels with increased numbers of WT T cells in OT1/OT2 Rag mice. Hence, our data reveals significant bystander T cell exhaustion independent from a tumor-antigen-T cell crosstalk.

However, the question remains as to how many CLL-specific T cells might be present within the pool of exhausted T cells in Tcl1 mice or CLL patients, and whether this unspecific exhaustion pattern together with other immune editing strategies create an overwhelmingly suppressive environment, which cannot be reverted by simple ICI therapy. This is hard to assess, as it is necessary to deduce specificities from the complete TCR composition [21]. Evidence that an adaptive anti-CLL response occurs in the Tcl1 mouse transplant model comes from the observation that CLL development is accelerated in the absence of a normal T cell repertoire [18], as also observed in our Rag-deficient and proficient OT1/OT2 recipient model in this study. In addition, the TCR repertoire is severely skewed in CLL, which indicates that antigen specificity plays a role for T cells during CLL development [11,18,22,23,24]. However, anti-CLL T cell responses could generally be rather weak and specific T cells could be silenced by mechanisms other than exhaustion, such as clonal deletion, conversion to regulatory T cells, or induction of anergy, senescence, or stemness [25]. Indeed, previous studies already confirmed higher Treg numbers in CLL patients compared to age-matched healthy volunteers [26]. Our data also suggest the increased susceptibility of CLL patients towards infections, as potential pathogen-specific cells from the naïve T cell repertoire could be silenced by the observed bystander exhaustion [27].

## 4. Materials and Methods

### 4.1. Mice

Experiments were conducted under the approval of the Austrian animal ethics committee (BMWF 66.012/0009-II/3b/2012 and TGV/52/11-2012, BMBWF-66.012/0002-V/3b/2018). CLL tumors were originally derived from the Tcl1 mouse model and further adoptively transferred into WT mice (C57BL/6 N/J), OT1 (C57BL/6-Tg(TcraTcrb) [15], OT2 (C.Cg-Tg(DO11.10)10Dlo/J) [28] mice, and OT1/OT2 mice harboring a deletion of the third exome of the Rag2 gene (OT1/OT2 Rag) (OT1 Rag: B6.Cg-Rag2tm1Fwa Tg(TcraTcrb)1100Mjb/Tac, MGI:4838647, OT2 Rag: B6.Cg-Rag2tm1Fwa Tg(TcraTcrb)425Cbn/Tac, MGI:4838759). All mouse models possess the same genetic background of C57BL/6 N/J. After sacrificing mice, spleen was removed and homogenized for flow cytometry analysis, BCR analysis or frozen for subsequent FACS sorting or transplantation. CLL tumors were transplanted by intraperitoneal injection of splenocytes including CLL cells. Injected cell numbers varied according to tumor and transplantation experiment: in the preceding two transplantation experiments 4 × 10^6^ splenocytes of tumor 658, 5 × 10^6^ splenocytes of tumor C755 and 1.425 × 106 splenocytes of tumor 703 were injected separately in WT, OT1, and OT2 mice (Figure 1A). For the subsequent transplantation experiment, CLL cells derived from the spleen of the WT mice 370 and 371 (transplanted in the previous experiment using tumor 703) were untouched. MACS purified (Pan B cell isolation Kit, 130-104-443, Miltenyi Biotech, Bergisch Gladbach, Germany) via an antibody cocktail (CD3ε, CD4, CD8a, CD49b, Gr-1, and terr119) targeting non-B/non-CLL cells and transplanted individually into WT, OT1, OT2, OT1 Rag, and OT2 Rag mice (tumor 370 1.725 × 10^6^ cells; tumor 371 1.6 × 10^6^ cells). Additionally, healthy B cells of the untransplanted WT mouse 483 were purified in the same manner and transplanted into OT1 Rag mice as a control (1.48 × 10^6^ cells) (Figure 3A). Tumor load was checked weekly via flow cytometry measurements of venous blood samples and mice were sacrificed by CO_2_ suffocation when showing signs of weakness or a lymphocytic tumor content of >80%.

### 4.2. Flow Cytometry

Tumor load and PD1 levels on T cells were measured on a Gallios and a CytoFLEX system (Beckman Coulter, Brea, CA, USA) using the following antibodies: anti-mouse CD19 FITC, clone: 6D5, cat: 115506; anti-mouse CD3 AF700, clone: 17A2, cat: 100216; anti-mouse CD4 APC, clone: RM4-5, cat: 100516; anti-mouse CD44 PC5.5, clone: IM7, cat: 103032, anti-mouse CD5 BV421, clone: 53–7.3, cat: 100629; anti-mouse CD19 BV605, clone: 6D5, cat: 115540; anti-mouse CD4 BV650, clone: RM4-5, cat: 100555; anti-mouse PD1 BV785, clone: 29F.1A12, cat: 135,225 (Biolegend, San Diego, CA, USA); anti-mouse Vβ5.1/5.2 PE, clone: RM2-5, cat: 562086; anti-mouse Vβ5.1/5.2 PE, clone: RM9-4, cat: 553190; anti-mouse CD8 APC-H7, clone: 53–6.7, cat: 53–6.7 (BD, Franklin Lakes, NJ, USA); anti-mouse CD5 PC5.5, clone: 53–7.3, cat: 45-0051-82; anti-mouse CD8a PO, clone: 5H10, cat: MCD0830; anti-mouse CD62L FITC, clone: MEL-14, cat: 11-0621-85 (Invitrogen, Carlsbad, CA, USA); anti-mouse PD1 eFluor 450, clone: RMP1-30, cat: 48-9981-82 (eBiosciences, San Diego, CA, USA). For flow cytometry of EDTA peripheral blood, 10–20 µL blood was stained in PBS (15 min at RT) and erythrocytes were lysed using a FACS lysis solution (BD, cat. 349202, Franklin Lakes, NJ, USA). Spleen samples were mashed and erythrocyte lysis was performed using ACK buffer (5 min at RT). Subsequently, cells were washed with PBS and stained in PBS (15 min at RT). Isotype control was used to define threshold for PD1 positivity. Percentage of PD1 expression is referred to CD3pos and/or CD5pos CD4pos/CD8pos Vb5pos/Vb5neg populations. TCR repertoires of spleen samples were analyzed using the mouse Vβ TCR Screening Panel (BD, cat. 557004, Franklin Lakes, NJ, USA).

### 4.3. BCR Analysis

DNA of 5 × 10^6^ splenocytes was isolated via the DNeasy Blood and Tissue Kit (Qiagen, Hilden, Germany). BCR sequencing libraries were prepared according to an in-house established protocol, as previously described [29]. Libraries were quality checked using the Tapestation Bioanalyzer (Agilent, Santa Clara, CA, USA) and sequenced (300 bp paired-end) on a MiSeq instrument (Illumina, San Diego, CA, USA). Sequencing reads were analyzed using the MiXCR software (version 2.1.8, MiLaboratory, Sunnyvale, CA, USA) in order to extract CDR3 sequences of IGH. BCR clones comprising of <1% clonal fraction, showing a quality score of <Q30 and/or display mutations within the CDR3 region were bioinformatically excluded from analysis.

### 4.4. Cell Sorting

Homogenized frozen spleen samples were sorted on an ARIA III instrument (BD, Franklin Lakes, NJ, USA) using the following antibodies: anti-mouse CD3 AF700, clone: 17A2, cat: 100216; anti-mouse CD19 PE/Cy7, clone: 6D5, cat. 115520; anti-mouse CD4 PE, clone: RM4-5, cat: 100,512 (Biolegend, San Diego, CA, USA); anti-mouse Vβ5.1/Vβ5.2 FITC, clone: MR9-4, cat: 51-01354L (BD, Franklin Lakes, NJ, USA); anti-mouse PD1 eFluor 450, clone: RMP1-30, cat: 48-9981-82; anti-mouse CD8 APC, clone: 53–6.7, cat: 17-0081-82 (eBiosciences, San Diego, CA, USA). Frozen OT1/OT2 splenocytes were thawed in RPMI medium (10% FCS, 1% pen/strep, 1% L-Glut) and filtered (30 µm, Sysmex, Goerlitz, Germany) before they were stained in sort buffer (PBS supplemented with 2% FCS and 0.025M HEPES) for 15 min at RT. Thereafter, cells were washed with sort buffer and sorted for CD4/CD8 Vb5pos/neg PD1pos/neg populations. Sorted T cell populations consisted of 1000 cells/population. Purity of sorted cell populations was checked regularly, and purity was found to be consistently high—T cell purity: 88.9–100%; CD4/CD8 purity: 88.6–99.7%; Vb5pos/neg PD1pos/neg populations: 97–100%. Different cell populations were sorted directly into RLT lysis buffer and RNA was immediately isolated after the sorting process using the RNeasy micro Kit (QIAGEN, Hilden, Germany).

### 4.5. RNA-Seq Analysis

Isolated RNA derived from sorted T cells (described above) was used for RNA-Seq library preparation using the NEBNext Single Cell/Low Input RNA Library Prep Kit for Illumina (NEB, Ipswich, MA, USA), which were subsequently quality checked using the Tapestation Bioanalyzer (Agilent, Santa Clara, CA, USA). RNA libraries were sequenced (75 bp single-end) on a NextSeq instrument (Illumina, San Diego, CA, USA). Individual libraries were covered by a mean of 20.6 × 10^6^ reads (SD = 3 × 10^6^). Integrity and sequencing quality of the output fastq files was evaluated using FastQC (v0.11.5, Babraham Bioinformatics, Cambridge, Great Britain) [30]. Reads were adapter- and quality-trimmed using Trimmomatic [31] (TruSeq3-SE adapters including polyA tail and default settings). Trimmed sequencing reads were aligned using the STAR aligner (v2.7.3a, Cold Spring Harbor Laboratory, New York, NY, USA) [32] and refGene-annotated mm10 genome. Read summarization to mm10 refGene exons was performed using featureCounts (subread-1.6.3 package, The Walter and Eliza Hall Institute of Medical Research, Parkville, VIC, Australia) [33], with a minimal overlap of 30 bases. Differential RNA expression analysis was calculated using the R package edgeR (v3.26.8, Garvan Institute of Medical Research, NSW, Australia) [34]. In brief, genes with no or low read counts were filtered (filterByExpr), a normalization factor was calculated for each sample (calcNormFactors) and dispersions across all tags was calculated (estimateDisp). The Genewise Negative Binomial Generalized Linear Models with Quasi-likelihood Tests (glmQLFit) was used to compute for differences between two groups. A table of normalized expression values was extracted (cpm, normalized.lib.sizes = T, prior.count = 2, log = T). Volcano plots were created using the R package “EnhancedVolcano” [35] and custom sets of genes were marked for labeling. Genes were ranked by -log10(FDR)/sign(logFC) and the “fgsea” R package [36] was used to plot gene set enrichment (fgseaMultilevel, eps = 0). Gene sets deregulated in CD4+ or CD8+ exhausted T cells were acquired from Crawford et al. [19]. Multiple testing correction algorithm “fdr” and a cutoff of 0.05 was used to determine statistical significance in all analyses. RNA sequencing data were deposited in Sequence Read Archive (SRA), NCBI, NIH (SRA accession code SUB9885102).

### 4.6. Statistical Analysis

Statistical significance of median survival difference and combined Kaplan–Meier survival curves were compared for trend using a Log-Rank (Mantel-Cox) test using GraphPad Prism (8.2.0). PD1 expression levels of mice were compared by a paired/unpaired Student’s *t* test in GraphPad Prism (8.2.0).

## 5. Conclusions

Summarizing, our data show that solely the frequencies of PD1pos T cells in the context of cancer are not sufficiently informative to assess a possible sensitivity towards ICI, because PD1 can be upregulated in T cells without specificity towards a particular chronic antigen or towards the malignant cell. Further research will be necessary so as to identify and track CLL-specific T cells alongside disease development in order to examine and improve the usability of ICI for treatment of CLL.

## Figures and Tables

**Figure 1 ijms-22-06648-f001:**
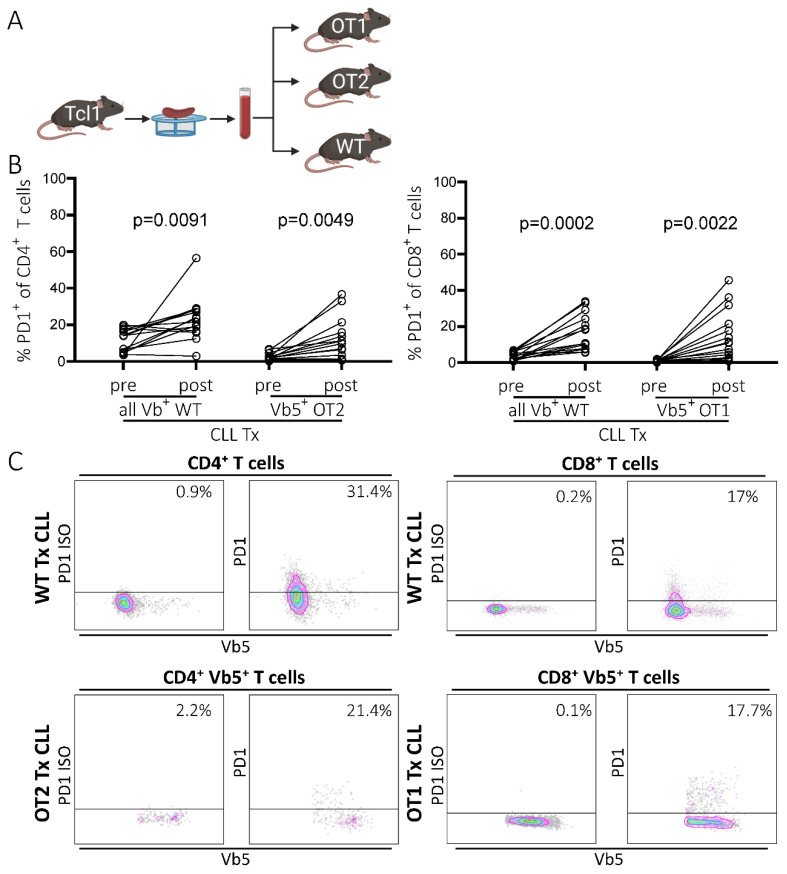
PD1 expression of T cells in WT and OT1/OT2 mice in peripheral blood, before and after CLL tumor transplantation. (**A**) Transplantation scheme. CLL tumors were derived from the Tcl1 mouse model. Splenocytes containing CLL cells were transferred intraperitoneally into congenic recipient mice (created with BioRender.com, 11 June 2021). (**B**) Percentage of PD1 positivity in the CD8/CD4 T cell compartments of WT mice (all Vbpos) and of transgenic Vb5pos CD8 (OT1 mice) and Vb5pos CD4 (OT2 mice) T cells in peripheral blood before and post-transplantation (last measurement before sacrifice day) analyzed by flow cytometry. (**C**) Representative flow cytometry data showing PD1pos T cell fractions in peripheral blood samples of WT and OT1/OT2 mice post-transplantation. Left panel shows PD1 isotype control, right panel shows actual PD1 signal.

**Figure 2 ijms-22-06648-f002:**
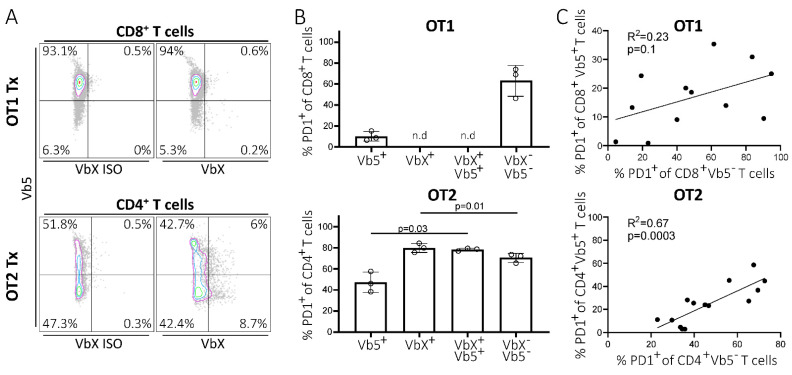
Non-cancer-specific T cells upregulate PD1 after CLL transplantation. (**A**) Representative flow cytometry data of splenocytes of transplanted OT1/OT2 mice stained with a FITC-conjugated antibody cocktail targeting various TCR Vb chains (VbX) and a PE-conjugated antibody recognizing the OVA-specific TCR (Vb5). The VbX antibody cocktail recognizes around 71% of all endogenous TCR Vb chains of a WT mouse. Thus, Vb5neg VbXneg populations represent endogenous T cells expressing single TCR Vb chains not covered by the VbX antibody cocktail. The left panel represents an isotype control for VbX, the right panel shows the actual VbX signal. (**B**) PD1pos T cell fractions of OT1/OT2 mice (*n* = 3), of the flow cytometry quadrants depicted in A. Populations consisting of <100 events were not analyzed. P values are only shown when significant. Error bars represent SD. (**C**) Correlation of PD1pos T cell fraction in Vb5pos and Vb5neg T cell populations in OT1/OT2 mice. n.d.: not detected, SD: standard deviation.

**Figure 3 ijms-22-06648-f003:**
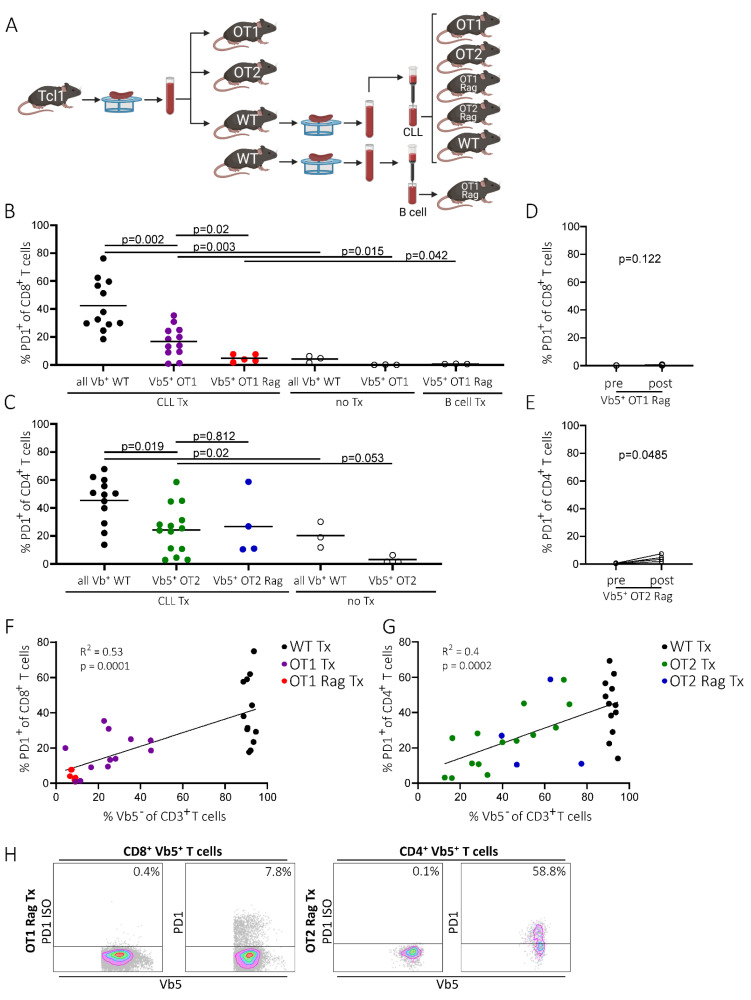
PD1 expression of T cells in OT1 Rag and OT2 Rag mice. (**A**) Transplantation scheme of OT1/OT2, OT1 Rag/OT2 Rag, and WT mice. Spleens derived from transplanted or non-transplanted WT mice were homogenized and T cells were depleted via a pan-B/CLL cell MACS isolation in order to inject CLL or healthy B cells without large amounts of endogenous T cells (created with BioRender.com, 11th June 2021). (**B**) Flow cytometry data of splenic PD1pos CD8pos T cells of WT (all Vbpos), OT1 (Vb5pos), and OT1 Rag (Vb5pos) mice either transplanted with CLL cells (CLL Tx), B cells (B cells Tx), or non-transplanted (no Tx). Bars represent mean. (**C**) Splenic CD4pos PD1pos T cells from either CLL-transplanted (CLL Tx) or non-transplanted (no Tx) WT (all Vbpos), OT2 (Vb5pos), or OT2 Rag (Vb5pos) mice. Bars represent mean. (**D**) Percentage of non-CLL-specific Vb5pos CD8pos PD1pos T cells in OT1 Rag mice in peripheral blood before and after tumor transplantation. (**E**) Percentage of CD4pos Vb5pos PD1pos T cells of OT2 Rag mice before and after tumor injection in peripheral blood. (**F**) Correlation of the amount of endogenous T cells (Vb5neg) (*x*-axis) with PD1 levels of CD8pos T cells (*y*-axis) in WT, OT1, and OT1 Rag mice. CD8pos T cells of *y*-axis correspond to endogenous T cells (all Vbpos) in WT mice and Vb5pos T cells in OT1 and OT1 Rag mice. (**G**) Correlation of the amount of endogenous T cells (Vb5neg) (*x*-axis) with PD1 levels of CD4pos T cells (*y*-axis) in WT, OT2, and OT2 Rag mice. CD4pos T cells of *y*-axis correspond to endogenous T cells (all Vbpos) in WT mice and Vb5pos T cells in OT2 and OT2 Rag mice. (**H**) Representative flow cytometry data of PD1pos spleen T cell (Vb5pos) fractions in OT1 Rag and OT2 Rag mice. Left panel shows isotype control of PD1, right panel shows actual PD1 levels.

**Figure 4 ijms-22-06648-f004:**
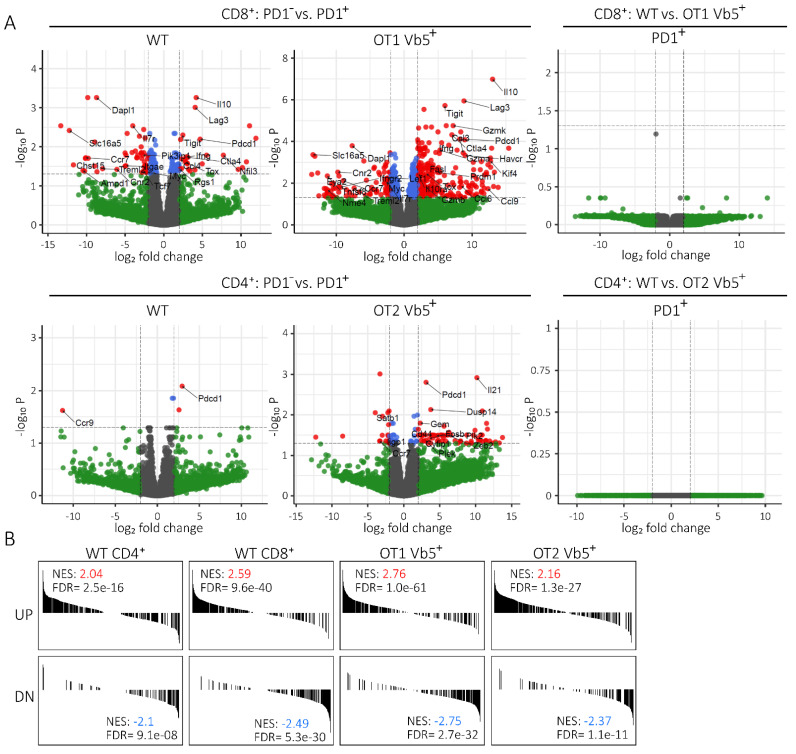
Gene expression profiling of sorted WT and OT Vb5-specific PD1pos T cells. (**A**) Differential gene expression analysis of PD1pos and PD1neg populations of CD8 (upper panel) and CD4 (lower panel), WT (all Vbpos) and OT1 or OT2 (Vb5pos) T cells. Log fold change (*x*-axis) is plotted against *p* values (*y*-axis) for each comparison. Genes upregulated or downregulated with a fold change ≥2 and an FDR <0.05 are depicted in red. Exemplary deregulated genes of the exhaustion gene sets defined by Crawford et al. [19] are annotated within the graphs. (**B**) GSEA of PD1pos versus PD1neg populations of CD4 and CD8 WT T cells and CD8 or CD4 Vb5pos T cells of OT1 and OT2 mice, respectively. CD4 and CD8-specific exhaustion gene sets defined by Crawford et al. were used for analysis [19]. Genes significantly upregulated in exhausted cells are shown in the upper panels and downregulated genes are shown below. UP: upregulated; DN: downregulated; GSEA: gene set enrichment analysis; NES: normalized enrichment score; FDR: false-positive detection rate.

## Data Availability

SRA accession code SUB9885102.

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
