# Peer review of "Evidence for Non-Cancer-Specific T Cell Exhaustion in the Tcl1 Mouse Model for Chronic Lymphocytic Leukemia"

_ijms, 2021, doi:10.3390/ijms22136648_

Round 1

Reviewer 1 Report

This manuscript details an interesting and important novel mechanism of immune exhaustion. In this model they clearly show that “bystander” immune exhaustion can be induced on non-CLL specific T cells that appears to be mediated through activated CLL-specific T cells. Overall, this study has been performed well. However, I do have a few comments:

  1. You state that OT1 and OT2 recipient mice show accelerated leukemia development (lines 90-93, Fig. S1B). Which percentage of CLL within all lymphocytes did you use to calculate this data and why (25%, 50%, end point)? Depending on which data points you use will probably greatly affect your statistical analysis.
  2. The conclusion sentence in the middle of paragraph 2.2 (lines 109-110) seems out of place and should be moved to the end of the paragraph to avoid confusion when reading this paragraph.
  3. The experiments with the Rag mice were performed with only 4 (OT2 Rag) or 5 (OT1 Rag) mice, whereas the previous experiments were performed with more than 10 mice per group. Particularly for measuring PD1 expression on non-CLL CD4 T cells in the OT2 Rag mice (Fig. 3C) the data could be affected by the use of low number of mice affect this data. How did you calculate how many mice to use in these experiments?
  4. Why is the transplantation with healthy B cells only performed in OT1 Rag mice and not in OT2 Rag mice or WT mice? If you want to observe whether CLL through activation of CLL-specific T cells induce the bystander PD1 upregulation on non-CLL T cells, should you not include transplantation of healthy B cells in WT mice as a control?
  5. You do not discuss why in OT2 Rag mice you do not see a further decrease in PD1 expression on non-CLL CD4 T cells (Fig. 3C), whereas you did observe the decrease on non-CLL CD8 T cells in OT1 Rag mice (Fig. 3B).
  6. In your discussion, you mention the “a large extent of these exhausted T cells could be non-CLL specific, hence, rendering their reactivation by ICI clinically ineffective.” (lines 285-286) Why would you want to reactivate non-CLL specific T cells in a therapy against CLL? They will still be ineffective against CLL. The bystander immune exhaustion of non-CLL specific T cells, however, will be important in the context of infections. You only dedicate one final sentence on this part, whereas this seems to be of great importance based on all your data.

Author Response

Reviewer 1:

You state that OT1 and OT2 recipient mice show accelerated leukemia development (lines 90-93, Fig. S1B). Which percentage of CLL within all lymphocytes did you use to calculate this data and why (25%, 50%, end point)? Depending on which data points you use will probably greatly affect your statistical analysis.

Thank you for this comment. As stated within the text (line 93) we used the assessment of 50% tumor load (TL) to calculate the tumor growth rate. The TL was measured weekly. The day post transplantation on which 50% TL was reached in each individual mouse was used to calculate the mean and subsequently the SD.  Overall survival of OT1 and OT2 mice also showed a reduced survival (OT1 31.1d, SD 6.7d, n=17, p=0.0001; OT2 37d, SD 8.1d, n=16, p=0.06; WT 42.7d, SD 7.7d, n=14) pointing to an accelerated tumor development.

We have chosen 50% tumor load as this value in our opinion is the time point, which separates tumor growths rates most robustly. This can be seen also in the presented mouse experiments in which tumor growth rates at 25% were still similar across all genotypes (OT1 26.7d, SD 7.9d, n=17; OT2 25.9d, SD 4.8d, n=16; OT1Rag 23.2d, SD 9.3d, n=5; OT2Rag 19d, SD 0, n=4; WT 24.5d, SD 6, n=14).

The conclusion sentence in the middle of paragraph 2.2 (lines 109-110) seems out of place and should be moved to the end of the paragraph to avoid confusion when reading this paragraph.

Thank you for this helpful comment. We also think that this conclusion sentence is far better at the end of this section and therefore moved it to line 126-127.

The experiments with the Rag mice were performed with only 4 (OT2 Rag) or 5 (OT1 Rag) mice, whereas the previous experiments were performed with more than 10 mice per group. Particularly for measuring PD1 expression on non-CLL CD4 T cells in the OT2 Rag mice (Fig. 3C) the data could be affected by the use of low number of mice affect this data. How did you calculate how many mice to use in these experiments?

Thank you for this important comment. The number of mice for OT1 Rag and OT2 Rag resulted from the availability of the mouse genotypes. Unfortunately, OT1 Rag and OT2 Rag mice do breed poorly and thus only 4 OT2 Rag and 5 OT1 Rag mice were available for the transplantation experiments. Of course, we would have preferred to use more mice as the results would be more robust. However, we think that this relatively low mouse number is sufficient to interpret the results since the individual mice display similar results.

Why is the transplantation with healthy B cells only performed in OT1 Rag mice and not in OT2 Rag mice or WT mice? If you want to observe whether CLL through activation of CLL-specific T cells induce the bystander PD1 upregulation on non-CLL T cells, should you not include transplantation of healthy B cells in WT mice as a control?

Healthy B cells were only transplanted in OT1 Rag mice because not enough OT2 Rag mice were available to perform this control experiment in both genotypes (as already mentioned in the previous paragraph). Naturally, we also would have preferred to conduct this control in both genotypes. We considered this control to be sufficient in OT1 Rag mice since we expect a potential influence of healthy B cells on OVA specific T cells to be independent on their CD4 or CD8 subtype.

Furthermore, transplantation of WT B cells into WT mice was not included into the experiment since we already obtained data in previous experiments which showed that the transplantation of healthy splenocytes into WT mice showed no significant change in PD1 expression on T cells (see figure in the attached word file).  

You do not discuss why in OT2 Rag mice you do not see a further decrease in PD1 expression on non-CLL CD4 T cells (Fig. 3C), whereas you did observe the decrease on non-CLL CD8 T cells in OT1 Rag mice (Fig. 3B).

Thank you for this comment. In line 198-206 we describe the correlation of higher fractions of endogenous T cells, which are present in OT2 Rag (comparable to conventional OT2 mice) and very low levels in OT1 Rag mice (Fig 3 F, G). We conclude that this higher endogenous T cell fraction in OT2 Rag mice, which most likely resulted from an outgrowth of tumor-specific T cells that were transplanted along with the tumor, is responsible for the T cell bystander effect of exhausted cancer-specific T cells on non-cancer-specific T cells. Since we did not observe a similar outgrowth of Vb5neg T cells in OT1 Rag mice, exhausted cancer-specific T cells, which could convey bystander exhaustion, are absent and thus we did not detect any PD1 upregulation on non-CLL-specific Vb5+ T cells.

In your discussion, you mention the “a large extent of these exhausted T cells could be non-CLL specific, hence, rendering their reactivation by ICI clinically ineffective.” (lines 285-286) Why would you want to reactivate non-CLL specific T cells in a therapy against CLL? They will still be ineffective against CLL. The bystander immune exhaustion of non-CLL specific T cells, however, will be important in the context of infections. You only dedicate one final sentence on this part, whereas this seems to be of great importance based on all your data.

Thank you for this comment. Tumors are often considered ICI sensitive solely based on their PD1 or PDL1 expression. T cell exhaustion is potently induced via repetitive TCR stimulation and thus exhausted T cells are often considered antigen-specific. The statement “a large extent of these exhausted T cells could be non-CLL specific, hence, rendering their reactivation by ICI clinically ineffective.” should point towards the possibility that a PD(L)1+ tumor, which is treated via PD1 targeting antibodies, might be insensitive towards this treatment because the PD1+ T cells are in fact not specific for any tumor antigen. We do not intend to treat non-cancer-specific PD1+ T cells via ICI but want to state that such a treatment might fail due to fact that it is unclear whether PD1+ T cells are tumor specific or not. In order to make this more clear we revised this sentence as follows: “An ICI treatment based solely on the presence of exhausted T cells might therefore unintentionally target primarily non-CLL T cells, hence, rendering any potential non-CLL T cell activation clinically ineffective.”(line 291-294)

Reviewer 2 Report

Major: In their submitted paper "Evidence for non-cancer-specific T cell exhaustion in the Tcl1 mouse model for chronic lymphocytic leukemia" the authors in Abstract claim that: "results indicate that in the CLL mouse model, a substantial fraction of non-CLL specific T cells becomes exhausted during disease progression in a bystander effect. These findings have important implications for the general usability of immune checkpoint therapies in CLL." However, in the part Conclusions their claim that "sole frequencies of PD1pos T cells in the context of cancer are not informative to assess a possible sensitivity towards ICI, because PD1 can be upregulated in T cells without specificity towards a particular chronic antigen or towards the malignant cell".

I as the reviewer am confused. One one hand "these findings have important implications for the general usability of immune checkpoint therapies (i.e. ICI) in CLL" on the other hand "frequencies of PD1pos T cells in the context of cancer are not informative to assess a possible sensitivity towards ICI, because PD1 can be upregulated in T cells.

Minor: The article is "full of data" for the reader it is not easy to follow all details. The authors should omit some experiments and pronounce more clearly the aim of the whole study.

Author Response

Reviewer 2:

Major: In their submitted paper "Evidence for non-cancer-specific T cell exhaustion in the Tcl1 mouse model for chronic lymphocytic leukemia" the authors in Abstract claim that: "results indicate that in the CLL mouse model, a substantial fraction of non-CLL specific T cells becomes exhausted during disease progression in a bystander effect. These findings have important implications for the general usability of immune checkpoint therapies in CLL." However, in the part Conclusions their claim that "sole frequencies of PD1pos T cells in the context of cancer are not informative to assess a possible sensitivity towards ICI, because PD1 can be upregulated in T cells without specificity towards a particular chronic antigen or towards the malignant cell".

I as the reviewer am confused. One one hand "these findings have important implications for the general usability of immune checkpoint therapies (i.e. ICI) in CLL" on the other hand "frequencies of PD1pos T cells in the context of cancer are not informative to assess a possible sensitivity towards ICI, because PD1 can be upregulated in T cells.

Thank you for your valued comment. We apologize for creating any confusion in our manuscript. Increased PD1 positivity within tumor samples is often assumed to be specifically expressed by tumor-specific T cells. PD1 expression, which is often used to assess ICI sensitivity, does not allow any conclusion concerning the specificity of PD1 positive T cells. The statement “frequencies of PD1pos T cells in the context of cancer are not informative to assess a possible sensitivity towards ICI, because PD1 can be upregulated in T cells without specificity towards a particular chronic antigen or towards the malignant cell” should note that ICI therapy in certain PD(L)1+ tumors might not be useful because of a lack of tumor specificity of exhausted T cells and should therefore be reevaluated. The sentence “These findings have important implications for the general usability of immune checkpoint therapies in CLL." should highlight that due to our novel finding of bystander exhaustion the sensitivity assessment of ICI therapies and therefore potentially the usability of ICI has to be reassessed. Further, our proposed bystander exhaustion might be partly responsible for PD1+ tumors to be insensitive to ICI therapies as observed for CLL.

In order to make these statements more clear we revised the following sentence (line 25): “These findings have important implications for the general efficacy assessment of immune checkpoint therapies in CLL.”

Minor: The article is "full of data" for the reader it is not easy to follow all details. The authors should omit some experiments and pronounce more clearly the aim of the whole study.

Thank you for this important comment. We included the following sentence into the manuscript and thereby hope to make the aim of the study more understandable to the reader (line 51-53): “Thus, the aim of this study is to elucidate whether T cell exhaustion is specifically confined to tumor-specific T cells or whether this dysfunctional phenotype also affects T cells independently of recognizing tumor antigens.

Concerning the amount of data presented within the main article we tried to include only the most important experiments and figures into the main text and provide any additional information in the supplements. Due to the substantial complexity of the study we think that the included figures should be presented within the main text in order to avoid any misunderstandings.